# A Bioinformatics Approach to Mine the Microbial Proteomic Profile of COVID-19 Mass Spectrometry Data

**Aziz Abdullah A. Alnakli, Amara Jabeen** , **Rajdeep Chakraborty, Abidali Mohamedali and Shoba Ranganathan \***

Faculty of Science and Engineering, Macquarie University, Sydney, NSW 2109, Australia;
aziz.alnakli@students.mq.edu.au (A.A.A.); amara.jabeen@mq.edu.au (A.J.);
rajdeep.chakraborty@hdr.mq.edu.au (R.C.); abidali.mohamedali@mq.edu.au (A.M.)
**\*** Correspondence: shoba.ranganathan@mq.edu.au

**Abstract:** Mass spectrometry (MS) is one of the key technologies used in proteomics. The majority of studies carried out using proteomics have focused on identifying proteins in biological samples such as human plasma to pin down prognostic or diagnostic biomarkers associated with particular conditions or diseases. This study aims to quantify microbial (viral and bacterial) proteins in healthy human plasma. MS data of healthy human plasma were searched against the complete proteomes of all available viruses and bacteria. With this baseline established, the same strategy was applied to characterize the metaproteomic profile of different SARS-CoV-2 disease stages in the plasma of patients. Two SARS-CoV-2 proteins were detected with a high confidence and could serve as the early markers of SARS-CoV-2 infection. The complete bacterial and viral protein content in SARS-CoV-2 samples was compared for the different disease stages. The number of viral proteins was found to increase significantly with the progression of the infection, at the expense of bacterial proteins. This strategy can be extended to aid in the development of early diagnostic tests for other infectious diseases based on the presence of microbial biomarkers in human plasma samples.

**Keywords:** mass spectrometry; severe acute respiratory syndrome coronavirus 2; COVID-19; metaproteome; trans-proteomics pipeline

## 1. Introduction

Proteomics is the comprehensive study of the whole set of proteins in a given organism, cell, or tissue. Of all the tools and technologies used to study proteins, mass spectrometry (MS) has been utilized most effectively primarily due to its ability to multiplex and obtain a high degree of confidence in the identification and quantification of proteins in a sample [1]. Coupled with advanced bioinformatics methodologies, MS is therefore the tool of choice for many researchers. The exponential growth in the use of MS has allowed for numerous advancements in the various fields of biology from medicine to horticulture, allowing the comparison of various variables in a single experiment (e.g., diseased vs. healthy) [2,3]. MS generates a tremendous amount of spectral information, which is stored as raw data in extensive public repositories such as PRIDE [4]. In most standard protocols, these data are searched against a specific database of interest to identify and quantify proteins from an organism. This approach, known as peptide mass fingerprinting (PMF), often results in a significant portion of 'unmatched' data being disregarded. In an alternative to PMF, several studies performed in the past decade have attempted to interrogate MS data using a de novo sequencing approach [5,6], with sequenced peptides searched again against defined databases. Both these approaches have their benefits, albeit with the fundamental assumption in most such studies that the majority of proteins in a sample come from the primary organism of interest (for instance, human) and hence pertain to a tightly defined search space.

It is well established that the human body has a substantial presence of microbes such as bacteria, viruses, and eukaryotic micro-organisms [7]. The number of bacterial cells in a

70 kg human body is estimated to be around 38 T, far exceeding the human cell count at approximately 30 T [8]. As a result, bacterial genes outnumber human genes 100-fold, with about 3 M genes exclusively present in the gut. Hence, the bacterial content is considered by some as the 'second human genome' [9]. Furthermore, bacteria account for about 2% of the total human mass, which, on average, equates to the size of the liver or the brain [10]. This is not particularly surprising when bearing in mind the long co-evolutionary history between human and bacteria [11]. In addition, the existence of bacteria can be either beneficial to humans (symbiotic), neutral (commensal relationship), or detrimental and therefore pathogenic. The study of this microbiota and their locale is tackled by an area of research called microbiomics [12]. The microbiome is defined as the sum of all genomes and genes of the members of microbiota, which is the collection of all the microbes present within a particular environment [13]. Disruption in the composition of normal microbiome leads to dysbiosis, which has been demonstrated to be a driver of some autoimmune and metabolic diseases [9]. The microbiome of each individual is specific in a complex yet analogous way to the genetic imprint, which may be a result of exposure to different environments before and after birth [13].

In some cases, microbes, especially viruses, can be detrimental and pathogenic upon their invasion of the host cells. Mammalian and bird cells are known to be reservoirs for non-pathogenic viruses that are harmless to the host cells. Once in the normal host, they are described as non-pathogenic and silent [14]. When cross-species transmission occurs, there is a chance that the virus turns becomes pathogenic within the new host. Severe Acute Respiratory Syndrome (SARS) is an example of a zoonotic disease where cross-species transmission led to the conversion of a silent virus to a pathogenic one [15]. SARS is caused by a virus belonging to the coronavirus genera. Studies have identified the ecology of the virus and how it might have transmitted to humans, suggesting that bats are the source of most of coronavirus cross-infections [16].

In addition, viruses have been integrated into the human genome at some early point in time. Retroviruses are also known to integrate their genetic material into the host genome as part of their virulence mechanism. Some ancient integrations, however, show beneficial effects on humans, where the products of their genes participate in key biochemical pathways. A study carried out by Chuong et al. [17] suggested that viral integration may be beneficial over time as viral proteins become part of the human protein network, playing an essential role in the defense process against subsequent viral infection. It is believed that the human immune system has evolved to make use of integrated viral genes and assist in the attack of pathogens. Chuong et al. deleted endogenous retroviruses by CRISPR-Cas9, resulting in an impairment in interferon-induced genes located in close proximity to the deletion site, which are crucial in immune response activation pathways, such as in the melanoma 2 (AIM2) inflammasome.

The early detection of microbial biomarkers can assist in the diagnosis of infectious diseases. For instance, testing for active tuberculosis is one of the areas needing improvement. While there are two diagnostic tests currently available for latent tuberculosis—namely, the intra-skin test (TST) and the IFN-$\gamma$ production test (IGRA)—both tests lack specificity, as they show positive results for subjects with prior bacillus Calmette–Guérin (BCG) vaccination and non-tubercular mycobacteria [18,19]. Furthermore, both methods are dependent on the detection of the immune response caused by the infection and not the causative bacterial agent.

The recent pandemic resulting from severe acute respiratory syndrome coronavirus 2 (SARS-CoV-2) [20] might be detected early in non-symptomatic patients with plasma peptides as biomarkers as opposed to rapid antigen tests that only detect peptides that are potent and plentiful in the nasal passages [21]. Non-infectious diseases have also been studied for the detection of microbial biomarkers. A study estimated that 15% of all human cancers are linked to viruses [22]. Therefore, it is quite plausible to have a microbial protein product indicating the presence of the microbial agent causing the disease. We

previously found that microbial antigens resulted in the proliferation of various cancer cells and lowered the efficacy of anti-cancer drugs [23,24].

The heterogeneous nature of biological samples, as exemplified above, is often incompatible with the narrow search parameters used in most proteomic studies. Our premise is that widening the search parameters (in this case, the search database) may lead to the uncovering of high-confidence peptides that can identify other non-host organisms of interest.

In this study, we developed a strategy to analyze raw MS data against the proteomes of bacteria and viruses present in healthy human plasma at a high confidence level (Figure 1) in order to obtain a more comprehensive view of the normal biological state. We then applied this strategy for the identification of bacterial and viral species as microbial biomarkers for the diagnosis of a latent infectious disease, SARS-CoV-2.

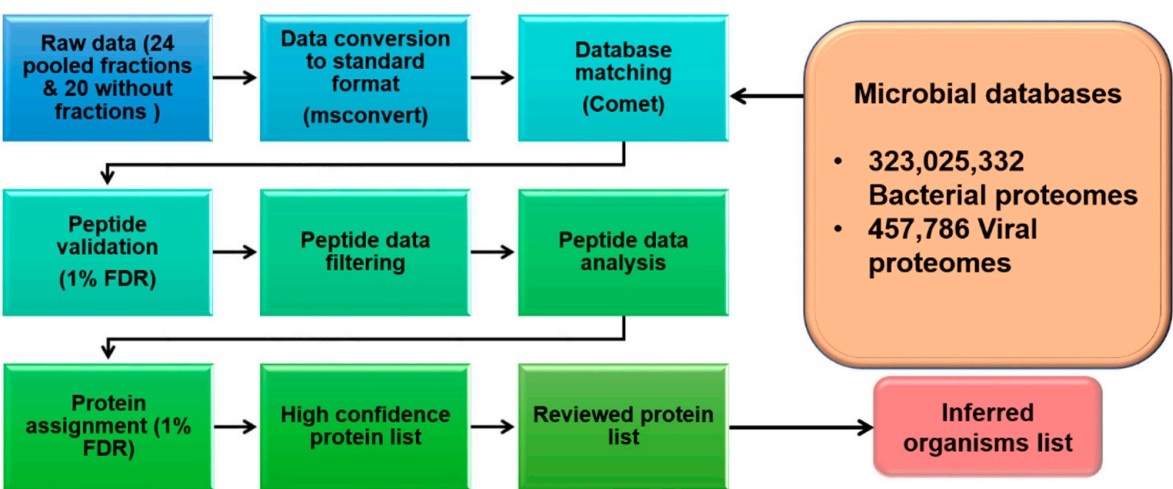

**Figure 1.** An overall chart of the major steps undertaken to identify microbial proteins in healthy human plasma.

## 2. Materials and Methods

### 2.1. Data Acquisition

This study targeted high-quality data from the mass spectrometric analysis of healthy human plasma (HHP) produced by Mann's group, which used Q-Exactive HF Orbitrap (Thermo Fisher Scientific Inc., USA) MS instrumentation, known for its high resolution and accuracy [25]. In this report, sample preparation was carried out according to the pipeline described in Geyer et al. [26]. Briefly, a subset of each plasma sample from 20 healthy individuals (10 males and 10 females) was depleted of the 20 most abundant proteins using two immunodepletion kits, as previously described [27]. After depletion, samples were pooled and then fractionated into 24 fractions using the high-pH reversed-phase "spider fractionation"[28] for deeper proteome coverage. MS data were obtained with a Top15 data-dependent MS/MS scan method [29]. Peptide identification by Mann's group was carried out with an initial precursor mass deviation of up to 7.5 ppm and a fragment mass deviation of 20 ppm. The accruing MS data were deposited in the PRIDE database [4], freely available via ProteomeXchange (http://www.proteomexchange.org, accessed on 1 February 2021), with the identifier PXD011749. The MS data, comprising 20 plasma samples from each participant and 24 fractions of the pooled plasma sample, were downloaded from ProteomeXchange in Thermo's RAW format.

The experimental design developed for HHP was tested regarding its efficiency in detecting microbial proteins by analyzing COVID-19 disease samples for SARS-CoV-2 proteins. SARS-CoV-2 proteomics MS data were collected from the proteomics database PRIDE [4] with the identifier UP000005640. Data were produced by Zhou's group in China in an effort to assign a protein biomarkers for SARS-CoV-2 in human plasma samples and

potentially aid in the early diagnosis and drug discovery of the disease [30]. The data were analyzed by Zhou's group at a 4.5 ppm mass accuracy for the precursor, with an initial search mass tolerance of 20 ppm, followed by a fragmentation mass tolerance of 20 ppm. Blood samples in this study was collected from 5 fatal, 7 severe, and 10 mild cases of SARS-CoV-2, portraying different stages of the disease. Additionally, healthy blood samples were taken from 8 subjects as a negative control. TMT labelling was performed with an internal reference included to allow the quantification of proteins. Each sample was fractionated into 60 fractions for each using high-pH reverse-phase fractionation; these were then combined into 20 fractions. LC-MS/MS data acquisition was performed using a Q Exactive HF-X mass spectrometer coupled with an Easy-nLC 1200 system. Since this dataset was from labelled peptides, the human plasma from Mann's group [25] could not be directly used for comparison, as the analysis procedure was slightly different. Plasma from fatal (FP), mild (MP), and healthy controls (HP) from Zhou's group were analyzed to find biomarkers for the early diagnosis of COVID-19.

## 2.2. Bacterial and Viral Protein Sequence Compilation

The available bacterial and viral proteome sequences were downloaded from UniProt [31] (as of 19 October 2020) in FASTA format. The compiled bacterial dataset had $3 \times 10^9$ protein sequences compared to the viral dataset of $7 \times 10^5$ protein sequences, originating from 7688 bacteria and 9484 viruses. This dataset was used to develop the microbial analysis platform for healthy human plasma (HHP).

Additionally, MS data from COVID-19 samples were also specifically searched against an updated viral dataset (20 February 2021) containing SARS-CoV-2 proteins. Additionally, to mitigate the loss of sensitivity arising from the use of such a small SARS-CoV-2 dataset, SARS-CoV-2 21 protein sequences were appended to human protein sequences, which were both downloaded from UniProt. The human proteome database (https://www.uniprot.org/proteomes/UP000005640, accessed on 20 February 2021) contained 20,366 reviewed human proteins, while the SARS-CoV-2 proteome database contained 21 proteins (ORF1ab (YP_009724389.1), ORF3a protein (YP_009724391.1), ORF6 protein (YP_009724394.1), ORF7a protein (YP_009724395.1), ORF7b (YP_009725318.1), ORF8 protein (YP_009724396.1), ORF10 protein (YP_009725255.1), nsp2 (YP_009742609.1), nsp3 (YP_009742610.1), nsp4 (YP_009742611.1), nsp6 (YP_009742613.1), nsp7 (YP_009742614.1), nsp8 (YP_009742615.1), nsp9 (YP_009742616.1), nsp10 (YP_009742617.1), nsp11(YP_009725312.1), leader protein (YP_009742608.1), surface glycoprotein (YP_009724390.1), membrane glycoprotein (YP_009724393.1), nucleocapsid phosphoprotein (YP_009724397.2) and 3C-like proteinase (YP_009742612.1)). The Comet parameter file was adjusted to include the parameters specified for the 11-plex TMT MS analysis [30] (http://comet-ms.sourceforge.net/notes/20171005_isotopiclabeling.php, accessed on 10 February 2021). The parameter file for the selected SARS-CoV-2 study was adjusted with the following changes: Nterm_peptide of 229.162932, K_lysine of 229.162932, and clear_mz_range between 125.5 and 131.5.

## 2.3. Data Processing through Trans Proteomic Pipeline (TPP)

The Trans-Proteomic Pipeline (TPP) (version 5.2.0) [32] was used to analyze the raw MS data against the bacterial and viral databases. Briefly, the msconvert module was used to convert the raw data to the standard mzML format. Database searching was performed through Comet using parameters provided on the comet website (http://comet-ms.sourceforge.net/parameters/parameters_201801/comet.params.high-high, accessed on 10 February 2021), which were adjusted based on the stringency required in ppm. The converted MS spectra were searched against the compiled bacterial and viral proteome datasets. Human plasma proteins were reported by Mann's group at 7.5 ppm [25], whereas Zhou's group reported human plasma from COVID-19 samples at 4.5 ppm [30]. To increase confidence in protein identification, healthy samples from Mann's study were investigated under three different mass accuracies—i.e., 5, 10, and 20 ppm—and only identifications common to the three mass windows were retained. A peptide-spectrum match (PSM) score

cut-off of 0.95 were applied, and data were filtered through PeptideProphet, retaining only peptides with high PSM values. The data processed by TPP were downloaded for local analysis and then compiled using a python script. The analysis strategy, comprising data processing and data analysis steps, is shown in Figure 1.

### 2.4. Analyzing the TPP Processed Data for High-Stringency Microbial Protein Identification

A multi-step workflow (Figure 2) was developed to retain only significant data based on highly stringent criteria. The first step involved the omission of all decoy sequences to maintain a false discovery rate (FDR) of less than 0.01 (i.e., <1%) [25,30]. In the second step, peptides with an identification probability (PSM) of more than 0.95 were retrieved and peptides equal to or more than 9 residues in length were retained for further analysis. Thirdly, non-proteotypic peptides were excluded from the analysis. A list of proteins identified with multiple peptide instances was compiled, and these were then categorized into reviewed and un-reviewed proteins by UniProt.

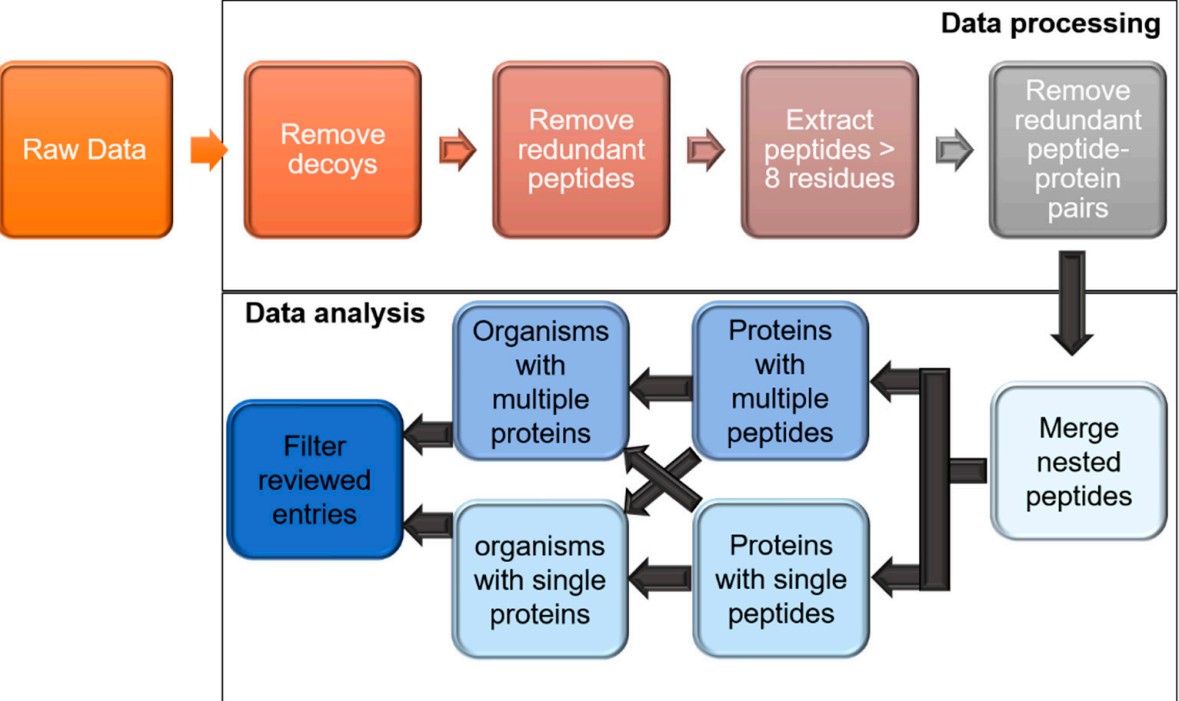

**Figure 2.** Workflow of data processing for output results from MS analysis, with data processing and data analysis sections.

### 2.5. Compilation of Bacterial and Viral Organisms

In the next step, the list of unique proteins was mapped to the organism source, as listed in the UniProt database. Next, a second classification step was applied, where organisms supported by a single protein as opposed to two or more proteins were distinguished to produce separate lists of organisms. Finally, only reviewed proteins listed by UniProt were retained, as these are considered as identified with strong experimental evidence compared to unreviewed proteins. The analysis steps are also summarized in Figure 2.

To monitor the microbial proteomic changes during the development of SARS-CoV-2 as opposed to the normal healthy microbial proteome of the human plasma, labelled MS proteomics data of plasma samples from fatal patients (FP), mild-disease patients (MP), and healthy controls (HP) were searched against the whole set of bacterial and viral proteins. To identify potential circulating bacterial and viral biomarkers for latent disease detection, the resulting lists of bacteria and viruses were compared separately in order to highlight

significant changes in the SARS-CoV-2 plasma samples vs. the healthy human plasma proteomic profiles.

## 3. Results

Microbial (bacterial and viral) proteins were identified in healthy human plasma in Mann's study [25] at 5, 10, and 20 ppm to ensure a high confidence in protein identification. These values were chosen to span the ppm values reported by the groups of Mann and Zhou. While the results obtained at 5 ppm are expected to be of the highest quality, it might be too stringent to identify all close matches, while 20 ppm may be able to identify many more peptides at a lower confidence level. To explore bacterial and viral presence in human plasma samples, we carried out identifications at three mass tolerance values to ensure that the metaproteome was captured accurately. We found that all proteins identified at 5 ppm were present in the 10 and 20 ppm results as well, confirming that no microbial proteins were lost by using the high-stringency cutoff of 5 ppm.

### 3.1. Overall Process of Bacterial Protein Filtration

Out of the 44 MS raw files, none of the unfractionated 20 files corresponding to each of the 20 participants in Mann's study [25] yielded any bacterial or viral peptide identification. The 24 pooled fractionated samples, on the other hand, were rich with identified peptides and, as a result, aided the discovery of the microbial proteome. Healthy human plasma samples were investigated for the existence of bacterial proteins using MS analysis. The MS spectra of healthy plasma were searched against the entire bacterial dataset, which contains over 323 million proteins originating from about 7700 bacterial strains.

The output results were filtered at a high PSM probability cut-off of 0.95 (5 ppm) in order to raise the confidence level of the peptide identification. This resulted in only 14,362 bacterial accession numbers (Acc. No.) surviving to this stage of the filtration process (Figure 3). The next step was to eliminate the non-unique mapping of peptides to proteins, in compliance with the Human Proteome Project (HPP) guidelines [33], as the same peptide could be present in different proteins. Therefore, peptides matching more than one accession number were omitted (412 peptides). Of the remaining proteins, only 18 proteins were identified with 2 or more non-nested peptides of at least 9 amino acid peptides, which aligns with the HPP high-stringency guidelines [33] (Figure 3).

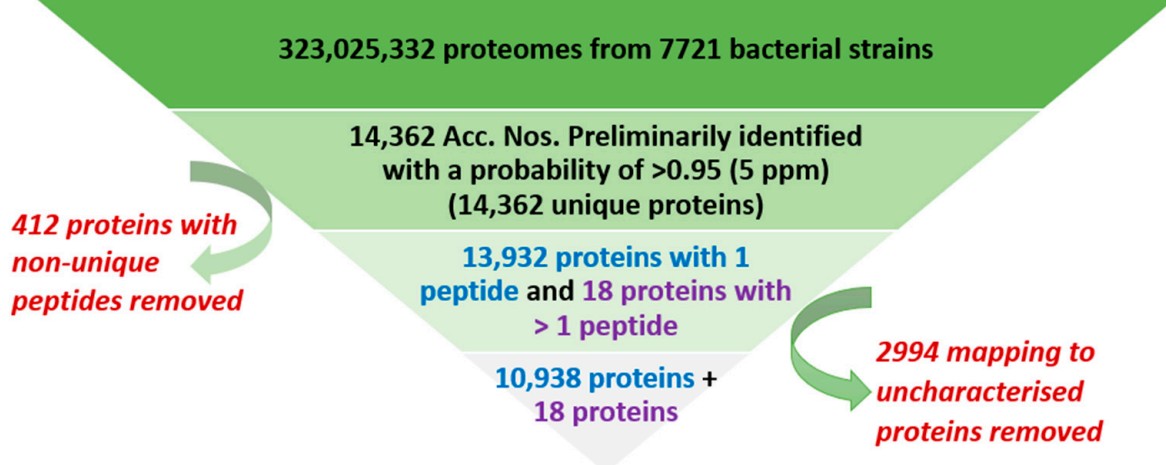

**Figure 3.** The overall flow chart for bacterial proteome analysis of healthy human plasma samples at a 5 ppm mass accuracy. A total of 323 M proteomes from 7.7 K strains were searched, resulting in the identification of 14,326 unique proteins. Matches to uncharacterized proteins were removed, resulting in the identification of 10,938 proteins identified with at least 1 peptide and 18 proteins with 2 or more peptides.

The rest of the proteins were identified by a single, unique, non-nested peptide of less than 9 aa peptides (Figure 3). Of those, only ~87% (10,938 with 1 peptide and 18 with >1 peptide) were functionally annotated and characterized, while the remaining (2994 peptides) were not yet annotated and labelled "uncharacterized" in UniProt. As a result, 18 bacterial proteins were identified with a 5 ppm mass accuracy (PSM = 0.95, as per the HPP stringency guidelines). None of the bacterial proteins found at this stringency were manually curated ("unreviewed") in UniProt, and were only computationally analyzed from the genome sequence of the organism (Supplementary Table S1). However, of the 10,938 proteins with 1 peptide, 61 were reviewed proteins.

### 3.2. Overall Process of Viral Protein Filtration in Healthy Human Plasma (HHP)

HHP samples were also investigated regarding the existence of viral proteins. The viral dataset contained over 457,786 proteomes originating from about 9515 viral strains. Data were processed using the same protocol as that used for bacteria (Figure 4). After passing the first processing stage, 1286 viral accession numbers (Acc. No.) were identified using a single unique peptide. Another 11 proteins were identified, in accordance with the HPP guidelines [33], with 2 or more non-nested peptides of at least 9 amino acids, all of which were 'reviewed' proteins in the uniport database (Figure 4).

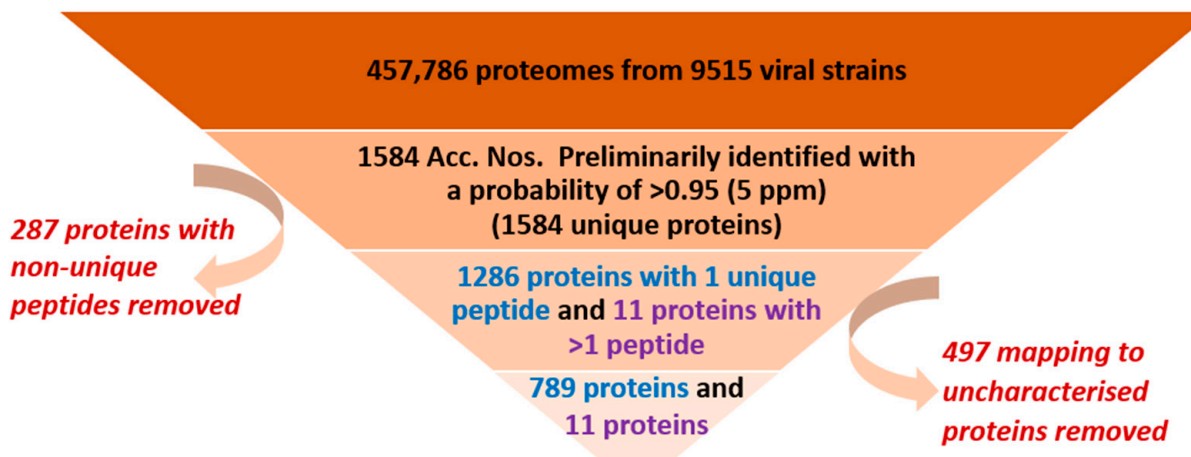

**Figure 4.** The overall flow chart for the viral proteome analysis of healthy human plasma samples at a 5 ppm mass accuracy. In total, 457 K proteomes from 9.5 K strains were searched, resulting in the identification of 1286 unique proteins. Matches to uncharacterized proteins were removed, resulting in the identification of 789 proteins identified with at least 1 peptide and 11 proteins with 2 or more peptides.

Supplementary Table S2 outlines the viral proteins that were detected in the healthy human plasma samples with the high-stringency HPP criteria. A couple of the proteins had even stronger evidence of being captured, as they were part of the manually annotated non-redundant protein sequence database (reviewed).

Another 43 reviewed viral proteins were identified through only a single peptide. Supplementary Table S3 lists the corresponding peptides and organisms.

### 3.3. SARS-CoV-2 Plasma Proteomics

To verify the existence of SARS-CoV-2 proteins in the human plasma of fatal (FP) and mild (MP) patients, the MS data from Zhou and coworkers [30] were searched against the UniProt complete viral and bacterial datasets using TPP (as described in Methods) and applying the same stringency criteria described for the healthy human plasma data from Mann's group. The SARS-CoV-2 metaproteome analysis was carried out at 5 ppm to ensure

the highest quality of results, with a high PSM probability cut-off of 0.95 (5 ppm) and an FDR of <1%.

Two peptides that mapped uniquely to two distinct SARS-CoV-2 proteins were found. The peptide GCCSCGSCCKFDEDDSEPVLK belonged specifically to a spike protein (a surface glycoprotein), while the peptide MADSNGTITVEELK was found to belong to a membrane protein. The proteotypic uniqueness of the peptides was independently validated by BLAST searches of the peptides through the NCBI server (https://blast.ncbi. nlm.nih.gov, accessed on 20 April 2021), reporting a very low E-value and a 100% coverage.

The complete bacterial and viral protein and organism content of healthy human plasma (HP) was compared with the plasma of fatal (FP) and mild (MP) patients using the MS data from Zhou and coworkers [30]. The results are shown in Figure 5.

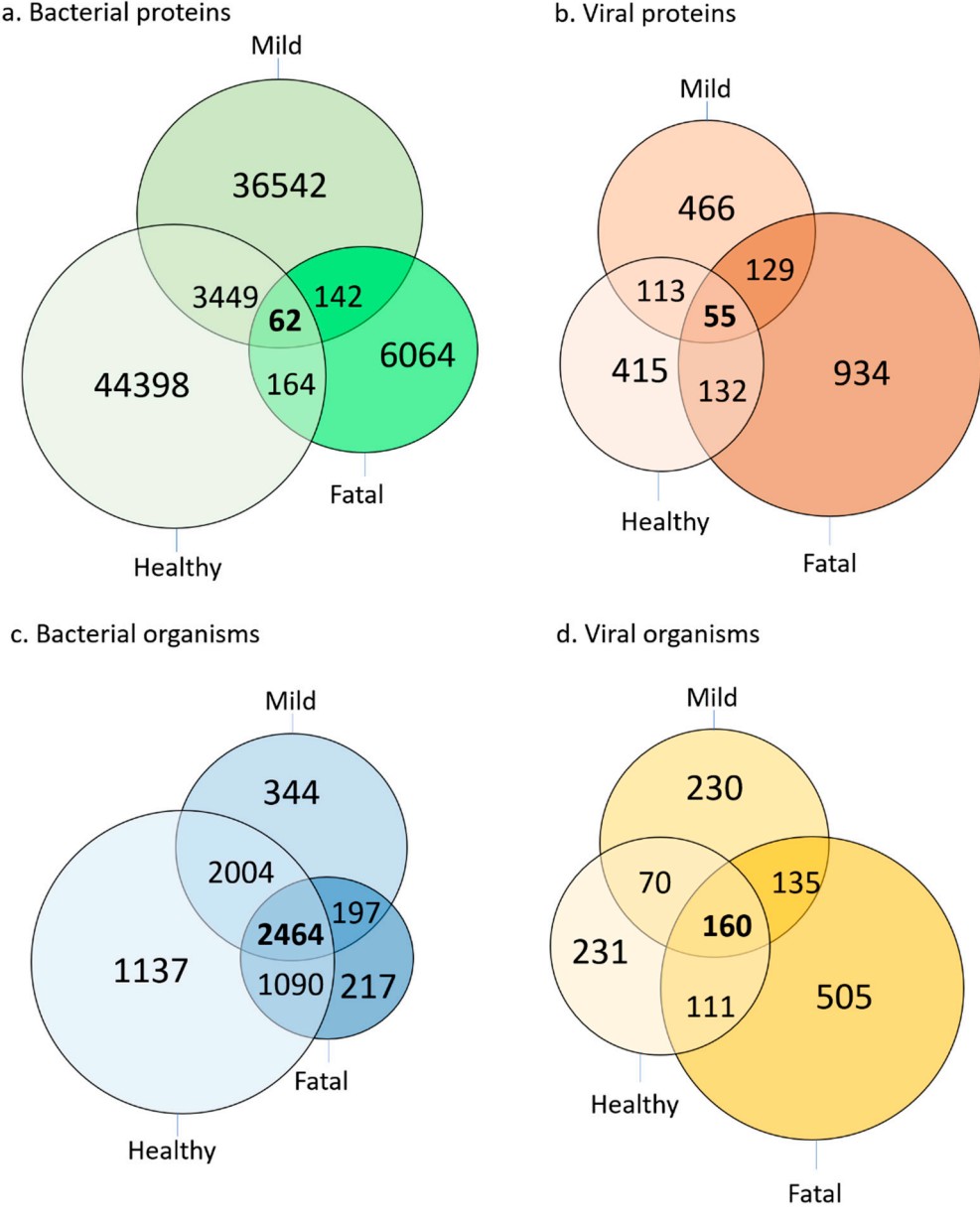

**Figure 5.** Protein profile and microbial distribution in the plasma of mild and fatal SARS-CoV-2 patients, compared to the plasma of healthy controls. Comparison of (**a**) identified bacterial proteins, (**b**) identified viral proteins, (**c**) bacterial organisms, and (**d**) viral organisms.

A similar pattern was detected when microbial proteins were assigned to their respective organisms. Of the 1442 viruses detected through MS, the plasma of fatal patients (FP) samples contained the highest number (505), followed by the plasma of healthy individuals (HP; 231) and the plasma of patients with mild symptoms (MP; 230).

Most bacteria, on the other hand, were detected at both stages of the disease (mild and fatal). The lowest number of bacteria is unique to FP samples, while HP samples are characterized by a decent collection of unique bacterial strains.

A one-way ANOVA test was applied to determine the number of proteins identified in FP, MP, and HP (Supplementary Table S4). Both bacterial and viral proteins were examined to investigate the degree of confidence by which they varied in quantity in each of the SARS-CoV-2 developmental stages. The test revealed a *p*-value of << 0.01, indicating a significant difference between the quantities of microbial proteins between the FP, MP, and HP samples. Additionally, this test demonstrates similar average quantities in each stage of the disease, with FP samples showing the most (~1.4) and HP samples showing the least (~0.6), along with MP samples with an average quantity of 0.85.

Overall, the mass spectrometry (MS) data of the FP (Supplementary Tables S5 and S6), MP (Supplementary Tables S7 and S8), and HP samples (Supplementary Tables S9 and S10) provide details of the viruses and bacteria in the identified proteins. As the disease worsens, it was found that the number of viruses in human plasma increases while that of bacteria decreases with respect to the number of bacteria in healthy human plasma.

## 4. Discussion

The study's primary aim was to investigate the presence of microbial protein remnants in healthy human plasma. This was accomplished through the reanalysis of high-quality publicly available MS data (primarily DIA/SWATH) of healthy human plasma samples, matched against the bacterial and viral databases. This approach is contrary to the conventional database searching approaches, where human samples are always searched against human databases This led to the development of a generic metaproteomic analysis approach applicable to any patient or model organism dataset for a particular disease.

Finding microbial proteins in human biological tissue is not new, and indeed many other approaches such as de novo sequencing have been attempted; however, these are fraught with unsurmountable statistical problems and therefore are of limited value [5,34]. Additionally, genomic studies are not as insightful as proteomic studies with regard to health and disease for biological samples. Until recently, concatenating and searching very large datasets (microbial) were not possible due to limitations in terms of compute power and software. Our results show that TPP can be used for comprehensive metaproteomic analysis.

Three mass accuracy parameters (5, 10, and 20 ppm) were used to reduce the possibility of errors in the database searching and acquire robustly identified proteins. The use of a mass accuracy of 5 ppm resulted in fewer than 10 or 20 ppm identifications, but all the proteins identified at 5 ppm were also present at 10 and 20 ppm. Therefore, for high-confidence identification, a 5 ppm deviation in mass accuracy was selected for analysis, with an FDR set to 1% for both peptides and proteins, as adopted by several high-quality studies [20,25,30]. Our study also prioritized the identification of reviewed proteins, which are manually curated and annotated by UniProt [31], from information in the literature and by computational analysis.

### 4.1. Confidence and Stringency of Identification

Mining microbial proteins using MS requires a set of guidelines to assure robust and reliable identification. The Human Proteome Project (HPP) classifies human proteins into classes according to the availability of scientific evidence. Proteins with the highest level of evidence are those observed experimentally at the protein level (protein evidence 1 or PE1). Other levels (PE2–PE5) encompass proteins inferred from the genome with no experimental verification of their protein product. Two non-nested and unique peptides

of at least nine amino acids, amongst other criteria, must be sighted by MS to qualify as a robust identification [5,34]. We have extended these high-stringency HPP metrics to identify the microbial proteins in healthy human plasma samples. Microbial proteins identified with less stringent yet accepted guidelines by the proteomics community are listed in Supplementary Table S3 for comparison.

### 4.2. Bacterial and Viral Proteins Identified in Healthy Human Plasma

The bacterial and viral proteins in HHP samples could be robustly identified through MS. More proteins were found for bacteria than for viruses, in line with the size of the corresponding genomes [34,35]. At the highest confidence level, 18 bacterial (Figure 3 and Supplementary Table S1) and 11 viral (Figure 4 and Supplementary Table S2) proteins were identified. Proteins identified at a lower confidence but acceptable level, as conventionally reported by the proteomic community, are listed in Supplementary Table S3.

It is worth noting that the proteins were only detected when the plasma samples of the 20 patients involved in the study were subjected to fractionation. As the samples were pooled and then fractionated, microbial proteins were detected. Several factors may contribute to the microbial proteins evading detection. One possible reason is the low abundance of such proteins, meaning that they fall under the detection limit of MS instruments [36,37], or that the proteins/peptides may simply be overwhelmed by highly abundant proteins (HAPs), a phenomenon commonly reported in plasma studies [38]. Another cause could be the unavailability of tryptic sites along the exposed regions of the protein sequences. This can be improved by using other digestion enzymes in combination with trypsin to maximize the probability of producing 'flyable' peptides detected by MS instruments [38,39].

### 4.3. Uncharacterized Microbial Proteins in Healthy Human Plasma

Uncharacterized proteins have no assigned function and are hypothesized to exist through their gene sequence, lacking identification at the protein level [40]. As most uncharacterized proteins are included in human and bacteria proteomes, a substantial number of uncharacterized proteins were identified in bacterial protein data [41]. About 21% of the bacterial and 40% of the viral proteins identified in this study have no annotated function. As MS is now the fundamental technology used to characterize and identify proteins, it can provide experimental evidence for a protein's presence [42]. This study uncovered evidence for 2994 bacterial and 497 hypothetical viral proteins based on a single unique peptide of 9 amino acids or more. Prioritizing these proteins for study over others with no protein level evidence is suggested in order to better understand their contribution to disease diagnosis and drug discovery by annotating their function.

### 4.4. Microbial Organisms Identified in Healthy Human Plasma

The mining of microbial proteins has allowed their viral and bacterial origins to be pinned down. This approach is becoming more commonly used, especially since the SARS-CoV-2 pandemic began in late 2019. This strategy was employed by Downard and coworkers [43], who detected viral peptides matching specifically to SARS-CoV-2 through proteomics. Similarly, two proteins related exclusively to SARS-CoV-2 peptides were captured by Zhou's group [30] to find human protein biomarkers in the plasma of fatal and severe cases. Our study discovered many microbial organisms in healthy human plasma.

The bacterial presence in healthy human plasma (HPP) was tracked through their proteins, leading to 151 bacterial organisms. Comparing the list to earlier studies in human blood microbiota, the results show many related phyla such as Proteobacteria, followed by Actinobacteria, Firmicutes, and Bacteroidetes [44,45]. HHP was dominated by Streptomyces and Clostridiales, amongst other bacterial species. The Clostridia class is thought to be dominant in the human plasma [46], which aligns with the findings reported here, as at least eight of the Clostridia strains were present in the list of organisms common to the three different mass accuracies.

Similarly, 24 viruses were confidently identified in HPP, with 25% being Pandoravirus strains. The virus was also seen in plasma through its DNA sequences by Colson et al. [47] and characterized as one of the giant viruses which have no known deterrent [48].

We also identified Yersinia phage, which was co-localized with the bacteria Yersinia. Yersinia enterocolitica is commonly known to cause transfusion-transmitted infection, as reported in the US [49] and New Zealand [50]. It is estimated that 46% of patients developing clinical sepsis after transfusion as a result of the contamination of transfused blood with Yersinia enterocolitica [51].

Human herpesvirus 7 and human herpesvirus 1 (Supplementary Table S3) were also observed in the healthy human samples. It is possible that latent diseases were not considered when assessing the health of the participants in this study. These herpes viruses are thought to infect a large percentage of the human population, with inactive herpes viruses becoming reactivated and resulting in severe pathological complications [52] when immunity is compromised.

Another virus that was found with high confidence was invertebrate iridescent virus type 3. The virus is tightly restricted to mosquitoes (Diptera) and is also called mosquito iridescent virus (MIV) [53]. It is very plausible that one of the participants in this study was bitten by a mosquito infected with this virus.

### 4.5. Case Study on SARS-CoV-2

Human plasma MS data samples from SARS-CoV-2 patients who did not survive the disease are already known to have a detectable amount of viral proteins [30]. Our analysis identified several SARS-CoV-2 proteins, with two unique SARS-CoV-2 peptides detected in the FP samples. The first peptide (MADSNGTITVEELK), which had a probability match of >95% to a membrane protein, was also reported by Downard's group [43], who investigated the presence of SARS-CoV-2 proteins in nasal swabs. This peptide could be a potential signature for this disease. The second peptide identified in this study (GCC-SCGSCCKFDEDDSEPVLK) belongs specifically to the spike protein (a surface glycoprotein) and appears to be a novel finding, as it has not been reported by other proteomics studies. Thus, these two peptides can potentially serve as biomarkers for the early detection of SARS-CoV-2 through blood tests, and also as candidates for vaccine development.

### 4.6. Bacterial and Viral Proteins Identified in SARS-CoV-2 Patient and Healthy Control Plasma

The viral proteome investigation proved its usefulness in distinguishing between the different developmental stages of SARS-CoV-2, as only around 2% of the proteins were common in HP, FP, and MP samples. A significant remaining proportion (81%) of the viral proteome appears to associate exclusively with one of the other three stages. These proteins are thus potential indicators for the stage of SARS-CoV-2 infection development. However, it is very noticeable that FP samples have the largest portion of the viral proteome. As SARS-CoV-2 endures, it exhausts the patient's immunity, resulting in a compromised immune system [54]. Therefore, more microbes are expected to invade the human tissues, some of which become detrimental and pathogenic as SARS-CoV-2 progresses.

### 4.7. The Microbial Proteome Changes in COVID-19 Samples Compared to Healthy

There is a noticeable change in the viral and bacterial proteome of the diseased samples as opposed to the healthy plasma samples. We observed a 42% increase in the number of microbial proteins detected in fatal patients (FP) compared to those with mild disease (MP) or healthy individuals (HP, Figure 5).

Information on the organisms from which these proteins arise is also useful in constructing a microbial organism profile for each developmental stage of SARS-CoV-2. As illustrated in Figure 5, each stage appears to be linked to a subset of viruses in human plasma. FP samples show a 35% enrichment of unique viral organisms. In comparison, both MP and HP samples have about 16% of the total viral count.

At all the stages of the disease (HP, MP, and FP), most of the corresponding proteome of samples is unique to that particular state. This opens the possibility that at least a subset of the proteome preliminarily linked to each stage can serve as a marker characterizing the severity of the disease. The established signature can either be specific to the disease and serve as a distinct indicator or be sensitive to the disease stage.

Future experiments should focus on testing the specificity of the method by determining the trend of other infectious diseases. It would be exciting to investigate whether a similar pattern can be discovered in other infectious diseases, both bacterial and viral. Additionally, as this study showed an increase in viral proteins stemming from the more significant number of viruses in FP compared to in MP and HP samples while the reverse was observed in the number of bacterial proteins and their bacteria of origin, it would be interesting to examine if a reverse trend can be observed in the case of bacterial infection.

### 4.8. Limitations

It is plausible that the microbial contamination of plasma samples during blood collection may have occurred. In fact, blood samples freshly collected even under a sterile environment are expected to contain a few bacterial spores. However, depending on the speed of the sample handling and the storage temperature, bacteria can pass the initial lag phase, allowing the initiation of exponential growth. Therefore, we recommend implementing more stringent experimental designs to reduce contamination and technical errors for future studies. Protocols with which blood is withdrawn and processed should also be improved to allow accurate plasma microbial studies. A strategy could be to study the microbial proteomics at the site of the puncture and those residing in the blood.

In addition, the quality of the results could be further improved by using more biological replicates in the experiment. Several MS data points from healthy human plasma samples produced in different laboratories should have been included. After that, only mutual proteins between all the replicates would be correlated to the HP samples.

## 5. Conclusions

This study comprehensively investigated the proteome of healthy human plasma for proteins derived from bacteria and viruses. MS analysis confidently proved the presence of several bacterial and viral proteins in healthy human plasma, some of which have already been verified at the genome level. The study emphasizes the significance of identifying microbial proteins in human samples, as they could aid in the early diagnosis of infectious diseases. Additionally, this is the first investigation, to the best of our knowledge, of the use of SARS-CoV-2 plasma to detect bacteria and virus changes at the protein and organism level as opposed to the use of healthy plasma; this is an approach that can be used to diagnose disease using peptides from the virus itself. Furthermore, two peptides that uniquely map to SARS-CoV-2 were detected, one of which was also reported in the nasal swabs of patients.

In this study, only human plasma samples were investigated. This allowed us to obtain a high-confidence and consistent microbial protein identification, as other biofluids, such as blood, mucous, saliva, urine, and feces, are highly dynamic and there could be wide variations in the results obtained. However, the approach developed here is generic and can be further extended and applied to other human biofluids. Additionally, more microbial proteins are expected to be present in such human biofluid samples, as they are more septic than plasma and the sample size will need to be much larger to be statistically significant.

The scope of this study can be further widened to include a more comprehensive set of human plasma samples. Additional high-resolution MS studies on healthy human plasma samples would be able to comprehensively identify the microbial species present in a large cohort, establishing a baseline for analyzing disease samples. This would allow the determination of the degree of microbial proteome heterogeneity between samples. Additionally, knowledge of the bacterial presence in SARS-CoV-2 patient samples might provide treatment options, especially for those requiring intensive care.

Furthermore, future research should focus on eliminating some of the challenges associated with the identification of microbial proteins in human samples. This includes the elimination of usual contaminants of microbial origin associated with the investigated sample type. This can assist us in narrowing the focus to only microbial proteins that are indicative of an individual's health status.

**Supplementary Materials:** The following supporting information can be downloaded at: https://www.mdpi.com/article/10.3390/applmicrobiol2010010/s1, Table S1: Unreviewed bacterial proteins identified through 2 or more peptides at 5 ppm in healthy human plasma (HHP); Table S2: Reviewed and unreviewed viral proteins found at high stringency at 5 ppm accuracy in healthy human plasma (HHP); Table S3: Reviewed viral proteins identified with a single peptide of at least 9 amino acids; Table S4: Statistical analysis of the number of proteins identified in the SARS-CoV-2 fatal patient plasma (FP), mild COVID-19 patient plasma (MP) and healthy plasma samples (HP); Supplementary Tables S5–S10 are available at https://doi.org/10.5281/zenodo.5812022, Table S5: Bacterial protein MS analysis data for SARS-CoV-2 fatal patient plasma (FP) samples; Table S6: Viral protein MS analysis data for SARS-CoV-2 fatal patient plasma (FP) samples; Table S7: Bacterial protein MS analysis data for SARS-CoV-2 fatal patient plasma (FP) samples; Table S8: Viral protein MS analysis data for mild COVID-19 patient plasma (MP) samples; Table S9: Bacterial protein MS analysis data for healthy plasma (HP) samples; Table S10: Viral protein MS analysis data for healthy plasma (HP) samples.

**Author Contributions:** Conceptualization, A.M. and S.R.; methodology, A.A.A.A., A.J.; validation, A.A.A.A., R.C.; formal analysis, A.A.A.A., A.M., A.J., S.R.; investigation, A.A.A.A.; resources, A.J., S.R.; data curation, A.A.A.A.; writing—original draft preparation, A.A.A.A.; writing—review and editing, A.M., R.C., S.R.; supervision, S.R. All authors have read and agreed to the published version of the manuscript.

**Funding:** This research received no external funding.

**Institutional Review Board Statement:** Not applicable.

**Informed Consent Statement:** Not applicable.

**Data Availability Statement:** The proteomic analysis results of the healthy human plasma are available from the authors on request.

**Conflicts of Interest:** The authors declare that they have no competing interest.

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
