# Peer review of "A Bioinformatics Approach to Mine the Microbial Proteomic Profile of COVID-19 Mass Spectrometry Data"

_2673-8007, doi:10.3390/applmicrobiol2010010_

Round 1

Reviewer 1 Report

The manuscript “A bioinformatics approach to mine the microbial proteomic profile of COVID-19 mass spectrometry data”, from Alnakli et al., reanalyzed MS data of previous publish works in order to search human blood microbiome, both bacterial and viral species, based on bioinformatic mining to the identification of microbial proteins. This work is a sign of these new times in which the sharing of data with the possibility of using it with other perspectives and new bioinformatic tools allows new conclusions and clues for future work. The authors used an appropriate experimental design with the concern of using indications, such as those of HUPO, in order to get stronger confidence on obtained data. The authors present a good exercise of raw MS data analysis in proteomics however the biological significance of the findings is not so evident. My major concern is the question of confidence in the identifications made, given that although there is reference to a verification of common access numbers and their elimination, BLAST searches are only mentioned in the second work in patients infected with SARS-Cov 2. My suggestion is that these aspects should be further clarified.

Author Response

The authors would like to thank the anonymous reviewers for their feedback and valuable comments, ultimately allowing for a publication of higher quality. We have gone through each individual comment and now revised the manuscript to address their concerns. The rationale behind each change has been included here. In the proposed publication, we have highlighted the revisions in different colours to allow for ease in tracking the changes made.

Changes based on Reviewer 1’s comments have been highlighted in blue.

Changes based on Reviewer 2’s comments have been highlighted in red.

Reviewer 2 Report

The presented study deals with up-to-date issue - COVID-19. The proper microbial proteomic analysis and comparison between different stages of the disease may have big influence on development of diagnostic tests, and perhaps development of new treatment strategies. The manuscript is very interesting. However, the discussion is very long and, in my opinion the most important findings are not properly highlight. Maybe the authors should consider minor changes and prepare the discussion more readable. Except for that small remark, I recommend this paper for publication.

Author Response

(The authors gave the same response as above.)
